# An Assessment of Vegetation Changes in the Three-River Headwaters Region, China: Integrating NDVI and Its Spatial Heterogeneity

**DOI:** 10.3390/plants13192814

**Published:** 2024-10-08

**Authors:** Xuejie Mou, Huixia Chai, Cheng Duan, Yao Feng, Xiahui Wang

**Affiliations:** 1Institute of Ecological Protection and Restoration Planning/Yellow River Ecology and Environment Protection Center, Chinese Academy of Environmental Planning, Beijing 100041, China; mouxj@caep.org.cn (X.M.); chaihx@caep.org.cn (H.C.); duanc.16b@igsnrr.ac.cn (C.D.); 2Key Laboratory of Water Cycle and Related Land Surface Processes, Institute of Geographic Sciences and Natural Resources Research, Chinese Academy of Sciences, Beijing 100101, China

**Keywords:** spatial heterogeneity, driving factors, random forest, Three-River Headwaters Region

## Abstract

Assessing vegetation changes in alpine arid and fragile ecosystems is imperative for informed ecological restoration initiatives and adaptive ecosystem management. Previous studies primarily employed the Normalized Difference Vegetation Index (NDVI) to reveal vegetation dynamics, ignoring the spatial heterogeneity alterations caused by bare soil. In this study, we used a comprehensive analysis of NDVI and its spatial heterogeneity to examine the vegetation changes across the Three-River Headwaters Region (TRHR) over the past two decades. A random forest model was used to elucidate the underlying causes of these changes. We found that between 2000 and 2022, 9.4% of the regions exhibited significant changes in both NDVI and its spatial heterogeneity. These regions were categorized into six distinct types of vegetation change: improving conditions (62.1%), regrowing conditions (11.0%), slight degradation (16.2%), medium degradation (8.4%), severe degradation (2.0%), and desertification (0.3%). In comparison with steppe regions, meadows showed a greater proportion of improved conditions and medium degradation, whereas steppes had more instances of regrowth and slight degradation. Climate variables are the dominant factors that caused vegetation changes, with contributions to NDVI and spatial heterogeneity reaching 68.9% and 73.2%, respectively. Temperature is the primary driver of vegetation dynamics across the different types of change, with a more pronounced impact in meadows. In severely degraded steppe and meadow regions, grazing intensity emerged as the predominant driver of NDVI change, with an importance value exceeding 0.50. Notably, as degradation progressed from slight to severe, the significance of this factor correspondingly increased. Our findings can provide effective information for guiding the implementation of ecological restoration projects and the sustainable management of alpine arid ecosystems.

## 1. Introduction

Since the mid-20th century, climate change and intensified human activities have led to the continuous degradation of ecosystems [1,2], particularly sensitive and fragile arid ecosystems [3]. Characterized by a complex mosaic of vegetation and soil patches [4,5,6], degradation in these ecosystems typically goes through several stages. At the early stage of degradation, vegetation is fragmented and interspersed with small patches of bare soil patches; as degradation progresses, small patches of bare soil expand into larger ones, eventually leading to desertification [7,8,9]. The most evident characteristic of this process is the decline in vegetation coverage, with vegetated areas gradually diminishing and becoming increasingly fragmented. Concurrently, bare soil areas, which initially are intermingled with vegetation, expand and eventually form continuous expanses. This results in discontinuous vegetation coverage in space, i.e., an increase in spatial heterogeneity [8]. Conversely, when arid ecosystems undergo recovery and improvement, vegetation regrows and gradually forms continuous patches, along with a reduction or even disappearance of bare soil. This leads to an increase in vegetation coverage and results in more homogeneous landscapes. By monitoring and analyzing the changes in the aforementioned indicators through field observations [3,9,10,11] or remote sensing [8,12], it is possible to effectively identify the degradation of arid ecosystems and provide early warnings for desertification [3,8,9,13,14,15], even facilitating timely ecosystem conservation and management measures.

The Three-River Headwaters Region (TRHR), which is the source of three major Asian rivers and is characterized by a typical alpine arid and fragile ecosystem in northwest China, has also experienced grassland degradation since the mid-to-late 1970s. This degradation process has continued to occur in recent years [16], with “black soil patches” in the degraded grasslands being very common [8,17]. Owing to its distinctive and critical ecological status, this region has consistently gained extensive attention from the scholarly community across the globe. So there have been numerous studies on ecosystem changes in the region, especially after the implementation of the Ecological Conservation and Restoration Project in the TRHR by the Chinese government. The majority of studies have reported on the changes in regional ecosystems since the implementation of ecological projects [18,19,20,21,22,23,24,25], such as assessing the effectiveness of ecological restoration by establishing comprehensive indicators that include ecosystem structure, quality, function, and influencing factors [18], or classifying the situation of grassland restoration in conjunction with remote sensing interpretation [19]; some studies further identified the contributions of different influencing factors such as climate and human activities after the analysis of vegetation changes [20,21,22,23,24,25]. However, these studies have primarily focused on changes in vegetation status or ecological function, with relatively less attention given to the fragmentation and spatial heterogeneity during the greening process of vegetation in the TRHR. Spatial heterogeneity is a key indicator for assessing the degradation of arid ecosystems [13]. For example, Li et al. [8] have established an integrated indicator for grassland degradation in the Qinghai–Tibet Plateau by combining changes in vegetation and its spatial heterogeneity, which has verified its effectiveness and provided a new perspective for our assessment of vegetation dynamics in TRHR. However, this study has not yet delved into the influencing factors of changes in NDVI and spatial heterogeneity, particularly the distinct impacts of various factors across different grassland types and regions experiencing varying vegetation dynamics.

In this study, our goal is to use a comprehensive indicator of NDVI and its spatial heterogeneity to assess the vegetation dynamics and their influence factors in the TRHR after the implementation of ecological projects. Specifically, we intend to achieve the following: (1) assess and meticulously classify vegetation changes by integrating the linear trends in NDVI and its spatial heterogeneity; (2) analyze the principal driving factors behind these new changes. Consequently, this study is aimed at providing a decision basis for the deployment of forthcoming regional ecological restoration projects and the sustainable management and enhancement of the region’s ecological integrity.

## 2. Materials and Methods

### 2.1. Study Area

Located in the eastern part of the Qinghai–Tibetan Plateau, the Three-River Headwaters Region (TRHR) is the birthplace of three major rivers in Asia: the Yangtze River, the Yellow River, and the Lancang River (Mekong River); it is therefore ecologically very important. The administrative district comprises 21 counties and 1 township, covering a total area of 384,000 km^2^. Within this area, meadows account for 55.76%, steppes for 24.05%, alpine vegetation for 9.4%, forests for 1.08%, shrubs for 5.41%, deserts for 0.54%, and other types for 3.75% [20], as depicted in Figure 1. Alpine meadows, with high vegetation cover and productivity [21], are primarily located in the central and southern parts of the study area. In contrast, steppes, with their relatively lower vegetation cover [22], are mainly situated in the western and northwestern regions. As the predominant ecosystem types in the TRHR [21,22,23,24], alpine meadows and steppes play crucial roles in maintaining biodiversity, carbon sequestration, and water conservation, which are essential ecological services [16,24]. However, these ecosystems are also highly vulnerable and sensitive to climate change and human disturbance activities [21,25]. The region is characterized by a quintessential plateau continental climate [20,26,27,28], with an average annual temperature of −17.5~9.6 °C, and an average annual precipitation of about 253.5~1191.2 mm over the past 20 years. It exhibits a northeast-to-southwest gradient, with colder, drier conditions in the northeast and warmer, wetter climates in the southwest (Appendix A). As a high-altitude region, TRHR features a topographical descent from northeast to southwest, with elevations ranging from 1970 to 6811 m (Appendix A).

### 2.2. Data Gathering and Processing

In large spatial scales, the Normalized Difference Vegetation Index (NDVI) satellite remote sensing data can be used to extract vegetation cover and its spatial heterogeneity. In this study, we used a monthly dataset of 250 m NDVI in China from 2000 to 2022 [29], which was generated based on the MOD13Q1 products of the Aqua/Terra-Moderate Resolution Imaging Spectroradiometer (MODIS) satellite sensor. The vegetation types data [30] were used for extracting the features of steppe and meadow types in the TRHR. Road network data obtained from OpenStreetMap [31] and river network data from the National Cryosphere Desert Data Center [32] were used to mask the higher spatial heterogeneity values caused by roads and rivers.

Meteorological data were derived from a high-resolution near-surface meteorological forcing dataset for the Third Pole region (TPMFD, 1979–2022) [33]; we calculated annual values of mean temperature, precipitation, and relative humidity based on the monthly data. SPEI-12 data came from high spatial resolution and century-long Standardized Precipitation Evapotranspiration Index (SPEI) datasets for China from 1901 to 2020 generated via machine learning [34], where the monthly values were also processed into annual values. Population density and actual livestock carrying capacity were taken as human activities factors. The population density data were obtained from the LandScan Global Population Dataset [35] (Oak Ridge National Laboratory, https://landscan.ornl.gov/), and the actual livestock carrying capacity primarily came from livestock carrying state estimation product in the Qinghai–Tibet Plateau [36]. Finally, all the driving factors were resampled to the same resolution.

### 2.3. Methods

#### 2.3.1. Spatial Heterogeneity and Its Changes

In this study, the spatial heterogeneity of NDVI was measured within a 3 × 3 pixel moving window of a 750 × 750 m area. The corresponding 3 × 3 pixel moving window provides 9 NDVI samples with a grid size of 250 m, which is statistically sufficient to calculate the coefficient of variation (CV). The spatial heterogeneity of the ecosystem is quantified by calculating the coefficient of variation within the selected moving window. The buffers of river and road networks were used to further cull unusually high heterogeneity values around these areas. Finally, impervious surfaces, waters, wetlands, and glaciers were masked out using land cover data [37].

The median NDVI and its CV within 3 × 3 pixel moving windows during the growing season (May–September) were calculated for each year from 2000 to 2022 of TRHR. The long-term changes in NDVI and spatial heterogeneity were measured by using linear regression, which is a common method [8,38], and areas with significant changes (*p* value < 0.05) were identified.

#### 2.3.2. Classification of Vegetation Changes

According to the definition of Li et al. [8], the combination of changes in NDVI and its spatial heterogeneity was used to classify the levels of vegetation changes (Table 1). Specifically,

(1) A combination of increases in NDVI and decreases in spatial heterogeneity represents improving conditions.

The healthy and intact grassland ecosystems are typically characterized by homogeneous landscapes with spatially consistent vegetation greenness. Consequently, an increase in NDVI signifies enhanced greenness and productivity within the ecosystem, and a decrease in spatial heterogeneity implies a more homogeneous vegetation distribution. The combination of these factors indicates an overall improvement in vegetation conditions.

(2) A combination of increases in NDVI and spatial heterogeneity represents regrowing conditions or slight degradation.

This combination can be divided into two different cases. In desert or sparsely vegetated areas with an average NDVI below 0.2 [39,40], an increase in NDVI signifies a likely transition from non-vegetative to low-vegetative conditions. This increase may occur in a small fraction of regions, resulting in spatial variation and increased spatial heterogeneity. Therefore, we interpret this combination in sparse or non-vegetated areas as regrowth of vegetation.

In the vegetated areas (NDVI > 0.2), an increase in invasive species may lead to a paradoxical situation where vegetation greening is accompanied by heightened spatial heterogeneity. This phenomenon occurs because invasive species, which often exhibit greater vegetation cover compared with the native species [6], can readily contribute to an increase in vegetation greenness. However, with the greening of invasive species, there may be fewer native species and more bare soil, thereby inducing the decline of homogeneity; that is, an enhancement of spatial heterogeneity. Consequently, these combined changes can be interpreted as initial or slight degradation.

(3) A combination of decreases in NDVI and increases in spatial heterogeneity indicates medium degradation.

In regions subject to fragmentation and degradation, increased anthropogenic pressures such as grazing may lead to an obvious decline in vegetation cover, with some local areas even experiencing exposure to bare soil. This can result in an increase in spatial heterogeneity, indicating a shift from a homogeneous to a heterogeneous landscape, thereby enhancing spatial heterogeneity. Consequently, this combination is indicative of a medium level of ecological degradation.

(4) A combination of decreases in NDVI and decreases in spatial heterogeneity represent severe degradation or desertification.

This can also be divided into two different cases. In vegetated areas with an average NDVI higher than 0.2, when severe landscape fragmentation and continuous degradation occur, there may be a significant reduction in NDVI and an increase in bare soil that becomes more extensive and connected. This is reflected in remote sensing data as a decrease in NDVI and a reduction in spatial heterogeneity, which indicates severe degradation.

In desert or sparsely vegetated areas (NDVI < 0.2), bare soil is already larger than that of vegetation. A decrease in NDVI indicates a further reduction and sparser vegetation, while conversely, the expansion of bare soil becomes more spatially homogeneous and extensive, leading to a decrease in spatial heterogeneity. These combined changes reveal a transition from the sparse vegetation stage to the bare soil stage, i.e., the desertification stage [8].

In conclusion, the levels of vegetation changes can be subdivided into six types, including desertification, severe degradation, medium degradation, slight degradation, regrowing, and improving conditions. In addition, according to the significance test results of a linear trend, only regions with significant changes (*p* value < 0.05) were further extracted.

#### 2.3.3. Driving Factors Analysis

In this study, Pearson correlation coefficients of NDVI and its spatial heterogeneity with individual influence factors were first calculated to explore the direction of effect caused by these factors. Then, the random forest model was employed to dissect the contribution of different factors that may lead to vegetation changes. NDVI and its spatial heterogeneity were taken as dependent variables, respectively; six influence factors including annual precipitation (Pre), annual mean temperature (Temp), relative humidity (RH), SPEI, grazing intensity (Graz), and population density (Pop) were used as independent variables. Specifically, over the past 20 years, the values of NDVI, spatial heterogeneity, and their influencing factors were extracted based on the grid points with significant vegetation changes. It should be noted that there are 12 types of grid points representing significant changes, which were obtained by integrating the maps of steppes and meadows with the six categories of significant vegetation changes delineated in Section 2.3.2. The samples of 12 types range from a few thousand to more than a million, and when the sample size of a certain type of area is too large, 10,000 samples are randomly selected for calculation, which is sufficient for a random forest model. Cross-validation was used to assess the impact of varying the number of trees on the model’s performance, thereby identifying the optimal quantity of trees within the model. Out-of-bag samples were utilized for model prediction to ensure the robustness and generalizability of the model’s predictive capabilities. The feature importance of various factors was assessed by calculating the mean decrease in impurity, a metric for the random forest model.

## 3. Results

### 3.1. Classification of Vegetation Changes by Integrating NDVI and Its Spatial Heterogeneity

#### 3.1.1. Linear Trend in NDVI

Over 2000–2022, 37.15% of the areas have seen a significant increasing trend in NDVI, whereas only 2.59% of the areas have experienced a decreasing trend, and no significant changed regions account for 60.3%. NDVI changes in different regions vary significantly; the regions that saw a significant increase are mainly distributed in the northeast, central, and northwest TRHR, such as Guinan, Xinghai, Zeku, Tongde, Maduo, and Qumalai, with a concentrated and continuous distribution. Regions that saw a significant decline are mainly dispersed in the central and southeastern TRHR, especially in the junction of Zhiduo and Qumalai and the junction of Gande, Dari, and Jiuzhi (Figure 2a). The changes in NDVI varied among different grassland types. The proportion of regions exhibiting significant greening is considerably higher within steppe areas (58.4%) compared to meadow regions (28.5%) (Figure 2b,c).

#### 3.1.2. Linear Trend in NDVI’s Spatial Heterogeneity

Between 2000 and 2022, the spatial heterogeneity of NDVI showed a significant increasing trend across 6.9% of the monitored regions. Conversely, a significant decrease was observed in 12.4% of the study area. The linear trends in NDVI spatial heterogeneity significantly differ from that of NDVI itself. The former is more spatially dispersed, with areas of significant increase primarily located in the western part of the study area, such as in the parts of Zhiduo and Tanggula. Areas that experienced a significant decrease are mainly found in the central and eastern parts of the region, such as in parts of Maduo, Dari, Chengduo, and Qumalai (Figure 3a). The variation in NDVI spatial heterogeneity among different grassland types is relatively minor. The percentage of regions with significant increases is marginally higher within steppe areas (10.6%) than that of meadow regions (5.6%) (Figure 3b,c).

#### 3.1.3. Refined Classification of Vegetation Improvement and Degradation

Based on the classification framework defined in Section 2.3.2, a refined classification of vegetation changes has been conducted over the past two decades. The findings reveal that approximately 36,096 square kilometers, accounting for 9.4% of the total change areas, are undergoing significant changes in both NDVI and its spatial heterogeneity. Among them, 62.1% exhibit a marked improvement in vegetation conditions, with a noticeable increase in vegetation and a reduction in spatial heterogeneity and fragmentation, predominantly in the eastern and northern sectors of the study area.

In sparsely vegetated regions with NDVI below 0.2, increases in both NDVI and its spatial heterogeneity suggest conditions of regrowth and recovery, which account for 11.0% of the areas exhibiting significant changes and are predominantly situated in the western of TRHR. Conversely, decreases in both NDVI and spatial heterogeneity, representing a mere 0.3% of the significantly changed areas, indicate a transition from a fragmented vegetation stage to bare soil, hinting at the risk of re-desertification.

Additionally, within vegetated areas where NDVI exceeds 0.2, a synchronous increase in NDVI and its spatial heterogeneity reveals slight degradation, approximately 16.2% of the significantly changed areas, and is mainly distributed in the western part of the study area. In contrast, the concurrent reduction in NDVI and spatial heterogeneity, representing 2% of areas exhibiting significant change, is indicative of severe degradation processes. Regions showed a significant decrease in NDVI and a significant increase in spatial heterogeneity accounting for 8.4%, which is indicative of medium degradation and primarily concentrated in parts of Zhiduo, Qumalai, Jiuzhi, Gande, and Dari (Figure 4a).

Further analysis was conducted in terms of different grassland types. The results showed that in contrast to steppes, meadows exhibit a higher proportion of areas with improved conditions and also contain areas where only medium degradation occurred; the specific figures are 50.7% vs. 69.4% (improving conditions) and 4.0% vs. 11.9% (medium degradation), respectively. Conversely, the proportions of areas with regrowing conditions and slight degradation in steppes are both higher than that in meadows; the figures are 22.1% vs. 2.3% (regrowing) and 21.9% vs. 13.5% (slight degradation), respectively (Figure 4b).

### 3.2. Variations and Contributions in Different Driving Factors

#### 3.2.1. The Varying Trends in Multiple Influence Factors

The slopes of variations in distinct driving factors are depicted in Figure 5. Over the past two decades, the TRHR has exhibited a pronounced trend towards a warmer and more humid climate. The statistical significance of linear trends indicates that an overwhelming majority of the regions (99.9%) have experienced an increase in average annual temperature, with 78.5% of these regions demonstrating a statistically significant rise (Figure 5b and Figure 6). In terms of annual precipitation, a substantial majority of the regions (93.7%) had an upward trend, with 28.5% of these regions showing a significant increase, predominantly in the southeastern and northwestern sectors of the study area (Figure 5a and Figure 6). Conversely, a significant proportion of the regions (85.3%) have observed a decline in Relative Humidity (RH), with 22.2% of these regions experiencing a statistically significant decrease, primarily in the western part of the study area (Figure 5c and Figure 6). Furthermore, SPEI has shown an upward trend in 74.6% of the regions, with 24.2% of these regions indicating a statistically significant increase, signifying enhanced humidity in the aforementioned areas (Figure 5d and Figure 6). The overall intensity of human activities within the study area has remained relatively stable with a modest decline. Specifically, regions with unchanged and decreasing population density account for 38.9% and 50.5%, respectively, while only 10.6% of the regions have experienced an increase in population density (Figure 5f and Figure 6). Grazing intensity has remained unchanged or decreased in 21.3% and 48.5% of the regions, respectively, with an increase observed in 30.2% of the regions, predominantly in counties where vegetation degradation is concentrated, such as Qumalai, Chengduo, Yushu, Tanggula, Zaduo, Nangqian, Gade, and Jiuzhi (Figure 5e and Figure 6).

#### 3.2.2. Correlative Analysis of NDVI and Its Spatial Heterogeneity with Influencing Factors

We calculated the correlation coefficients between NDVI, spatial heterogeneity, and different influencing factors. The results showed that NDVI exhibits significant positive correlations with climatic factors such as Pre, Temp, and SPEI, while it shows significant negative correlations with human activity factors like Graz and Pop. In particular, the areas with a significant correlation between NDVI and precipitation account for 40.1% of the study area, with 98.9% of these areas showing a positive correlation. Similar conclusions were drawn for the correlation between NDVI and temperature, as well as NDVI and SPEI, with significant correlation areas reaching 27.1% and 36.0%, respectively, and 94.6% and 99.2% of these areas exhibiting positive correlations. In contrast, the proportions of areas with a significant correlation between NDVI and human activity factors such as Graz and Pop are 24.6% and 16.6%, respectively, with 71.5% and 84.3% of these areas showing significant negative correlations.

The spatial heterogeneity of NDVI shows a significant correlation with each factor in a smaller proportion of areas compared to NDVI itself, and the directions of the impacts are opposite (except for RH). Specifically, it exhibits a predominantly significant negative correlation with the three climatic factors: Pre, Temp, and SPEI, with the proportions of the significant negative correlations being 82.0%, 61.7%, and 87.8%, respectively. In contrast, it shows a predominantly positive correlation with the two human activity factors, Graz and Pop, with the proportions of significant positive correlations being 50.4% and 66.0%, respectively. Regardless of whether it is the correlation between NDVI or the spatial heterogeneity of NDVI and RH, the proportions of areas with significant correlations are lower compared to other factors, being only 8.2% and 6.6%, respectively, with more than half of these showing significant negative correlations (Table 2).

Spatial distribution analysis reveals that areas with significant correlations between NDVI and Pre are predominantly located in the western, central, and northeastern parts of the study area (Appendix A). Regions where NDVI shows a significant correlation with Temp are mainly found in the western and southeastern areas (Appendix A). Areas with significant negative correlations with RH are primarily in the west, while those with significant positive correlations are mostly in the northeastern and southern central regions (Appendix A). Areas with significant correlations between NDVI and the SPEI are largely concentrated in the eastern part of the study area (Appendix A). Areas with significant negative correlations between NDVI and human activity factors such as Graz and Pop are predominantly in the northwestern, central, and northeastern regions (Appendix A). Conversely, the spatial heterogeneity of NDVI demonstrates a more scattered pattern of correlation with various factors across the study area, except for Pre and SPEI, which show denser clusters of significant correlations in the central and western regions (Appendix A).

#### 3.2.3. Feature Importance of Various Influence Factors to NDVI and Its Spatial Heterogeneity

The random forest model was employed to analyze the feature importance of various factors to vegetation dynamics in the 12 types of regions mentioned in Section 2.3.3 (Figure 7 and Appendix A). The results indicate that Temp and Graz are the most important factors affecting NDVI and the spatial heterogeneity of our study area. But the key determinants vary significantly across different grasslands and six types of vegetation changes. In terms of influencing factors of NDVI, in steppe areas with improving conditions of vegetation, Pre emerges as the predominant factor, ranking first in feature importance (Figure 7A and Appendix A). In steppe regions experiencing regrowing conditions, Temp is identified as the most influential factor, closely trailed by SPEI, Graz, and Pre, with minimal disparities in feature importance (Figure 7A and Appendix A). In steppe areas with slight degradation, Pre is the most crucial factor with a feature importance of 0.37, followed by Pop (Figure 7A and Appendix A). As degradation progresses from slight to medium and severe, the feature importance of Graz escalates, ultimately becoming the dominant factor in severely degraded areas, with an importance value of 0.56 (Figure 7A and Appendix A). In steppe areas at risk of desertification, Temp and Graz are the leading factors, with importance values of 0.39 and 0.22, respectively (Figure 7A and Appendix A). For meadow-type regions, distinct patterns are observed: Temp is the primary factor in areas with improving conditions and slight and medium degradation, with feature importance values of 0.55, 0.69, and 0.37, respectively (Figure 7B and Appendix A). SPEI and Graz are the main factors in meadow regions with regrowing conditions, with the same importance value of 0.23 (Figure 7B and Appendix A). In severely degraded meadows, Graz exerts the most substantial impact on NDVI, mirroring the patterns observed in steppe areas with severe degradation (Figure 7B and Appendix A). In meadow desertification zones, Graz and RH are identified as the most important factors (Figure 7B and Appendix A). In summary, the collective influence of climatic factors on NDVI in meadow areas, with an average importance of 0.74, surpasses that in steppe areas, which averages at 0.64.

The influence of factors on spatial heterogeneity in different regions diverges from those on NDVI. Temp consistently occupies the foremost position in terms of importance across various regions, except for the improving conditions and medium degraded zones in steppe ecosystems and the regrowing conditions, severe degradation, and desertification zones in meadow ecosystems (Figure 7C,D and Appendix A). In these five distinct zones mentioned above, aside from the steppe areas with improving conditions where Pre has the highest feature importance (Figure 7C and Appendix A), Graz is consistently recognized as the primary influencing factor in the other four types of areas (Figure 7C,D and Appendix A). Overall, the collective importance of climatic factors on spatial heterogeneity in steppe and meadow areas is similar, which averages at around 0.73.

Regardless of the types of vegetation changes, climatic factors have a higher feature importance on NDVI and its spatial heterogeneity, with average importance accounting for 68.9% and 73.2%, respectively, while the feature importance of human activity factors is less than one-third. The random forest model has demonstrated superior accuracy in NDVI simulation, yielding R-squared values mostly above 0.7. For the simulation of the spatial heterogeneity, the model’s R-squared values are modestly lower, ranging from 0.59 to 0.83, as depicted in the scatter plots presented in Appendix A.

## 4. Discussion

### 4.1. Integrate Spatial Heterogeneity into Vegetation Dynamics Assessment

Previous studies have shown that a negative NDVI trend indicates grassland degradation and a positive trend indicates grassland improvement [24,41]. Utilizing only the change trend in NDVI, existing research has demonstrated that more than 50–60% of the vegetation has been restored [24,41,42,43]. In comparison with studies focusing solely on the variation of NDVI, this study incorporates the spatial heterogeneity of NDVI into the vegetation dynamics assessment framework, allowing for a more nuanced reflection of the subtle vegetation changes. For instance, this approach can identify early stages of degradation within the process of vegetation greening, that is, areas where both NDVI and spatial heterogeneity increase significantly. The subtle degradation in these regions may be attributed to the proliferation of more invasive species (such as forbs), which tend to grow taller and have larger leaf areas compared to native species [8]. Our research reveals that in all areas with significant changes, approximately 16.2% exhibited a marked increase in both NDVI and its spatial heterogeneity during the period from 2000 to 2022, indicating the potential encroachment of invasive species leading to the fragmentation of grasslands.

Similarly, within areas exhibiting vegetation browning, the integration of increasing or decreasing spatial heterogeneity has enabled the further identification of regions experiencing medium and severe degradation, representing 8.4% and 2.0% of the areas with significant change, respectively. Furthermore, in sparsely vegetated deserts and bare lands (NDVI < 0.2), the concurrent increase in NDVI and spatial heterogeneity has uncovered instances of vegetation regrowth, accounting for 11.0% of the areas with notable change, indicating a clear recovery of vegetation; conversely, a joint decline in these indicators has pinpointed areas at risk of accelerated desertification, albeit at a minimal percentage of 0.3%. These new findings are crucial for the preservation and management of ecological stability in desert and bare land areas.

Although the study indicates that only a small fraction, 9.4% of the region shows significant changes in NDVI and spatial heterogeneity; this nuanced approach to analyzing vegetation dynamics effectively discerns the regional distribution of areas at various stages of degradation or recovery, which remains of significant importance for the future implementation of targeted ecological restoration projects and the management of ecosystem services.

### 4.2. Influencing Factors of Vegetation Changes and Suggestions for Ecosystem Management

Elucidating the principal determinants of vegetation variability is indispensable for shaping informed ecosystem management strategies. Many studies have consistently demonstrated that climate change and intensified human activities are usually the main causes of ecosystem changes [12,21,23,24,44,45]. Our investigation reinforces this consensus, revealing that most of the TRHR has experienced increasing trends in temperature and precipitation over the past two decades (Figure 5a,b), indicative of a clear warming and humidification trend [23]. However, certain local areas in the central and eastern parts of the study area are exhibiting an intensifying trend in grazing pressure (Figure 5e), which suggests that the impact of grazing activities is still prevalent. Overall, the impact of climatic factors on vegetation change in the TRHR substantially outweighs that of human activities, with a more than twofold difference in their importance, consistent with previous research [18,24,42,46].

The improvement of this study lies in its in-depth exploration of the differential dominant influencing factors within various types of grasslands and six finely classified vegetation change types. In terms of climatic factors, our findings further underscore the pivotal role of precipitation and particularly temperature as the primary climatic determinants of NDVI variability, which has been confirmed by previous studies [23,44]; the underlying mechanism of these factors is the pronounced elongation of the growing season [21,47,48,49]. Moreover, our findings reveal that climatic factors have a more pronounced impact on NDVI in meadow types, suggesting a greater susceptibility to climate change. However, the influence of these factors on spatial heterogeneity does not significantly differ between steppe and meadow types. Notably, Temp emerges as the primary driver of NDVI and its spatial heterogeneity across various regions of vegetation change, reaffirming its pivotal role in vegetation dynamics within the study area. Especially in the meadow type’s improving conditions and slightly degraded areas, the feature importance of Temp peaks at 0.55 and 0.69, respectively, indicating that the vegetation recovery in these areas is predominantly temperature-driven. This finding confirms the predominant role of natural climatic conditions in facilitating ecological restoration [50], suggesting a potential reassessment of human-centric restoration strategies. We recommend adopting more rational, nature-based solutions that capitalize on the favorable conditions provided by ‘climate assistance’ to expedite the recovery of alpine ecosystems.

Although the overall importance of human activity factors on NDVI is not pronounced in the study area, the impacts of grazing activities cannot be ignored, especially the considerable perturbations inflicted by localized grazing practices on the alpine arid ecosystem [44,51,52]. Overgrazing may cause and exacerbate grassland degradation [53]. Compared with previous studies, a novel finding of this study is the escalating influence of Graz on NDVI and spatial heterogeneity as steppes and meadows progress from slightly to severely degraded conditions, highlighting the substantial role of grazing in these degradation processes. For instance, in steppe and meadow areas with severe degradation, the impact of Graz on NDVI is particularly notable, with importance exceeding 0.50. These findings underscore the necessity for ongoing monitoring and strategic management interventions, such as the reduction in grazing intensity, to mitigate and prevent further degradation. However, it is crucial to acknowledge that human-driven ecological restoration measures often yield only a local direct impact, contributing only a small extent to an ecosystem’s recovery [54]. Especially in alpine arid ecosystems like the TRHR, where climate is a key driver, vegetation recovery and improvement on a large scale are predominantly influenced by natural factors. Thus, we advocate for an ecological restoration approach that considers the varying stages of vegetation change and applies targeted strategies. In regions showing significant improvement, natural restoration should prevail. For areas with degradation and desertification, a spectrum of proactive human interventions is necessary. Additionally, it is essential to foster conditions conducive to natural succession and recovery, thereby bolstering the resilience and stability of alpine ecosystems.

### 4.3. Limitation

In this study, we assessed spatial heterogeneity by employing a 3 × 3 moving window encompassing an area of 750 m by 750 m based on a pixel resolution of 250 m. This approach facilitates the detection of spatial structural variations in vegetation and bare soil coverage. However, detecting finer-scale spatial heterogeneity remains a formidable challenge. In addition, we did not consider the potential impacts of species composition and soil properties on the arid ecosystems in the study area, which may be different from the situation of ecosystem improvement or degradation in field investigations. And the analysis of the drivers of human activities still needs to be further refined. In the future, the above contents still need further supplementary research with field investigation data to provide an important decision-making basis for the restoration and management of the ecosystem in the study area.

## 5. Conclusions

In this study, we highlighted the vegetation dynamics within the TRHR by combining NDVI with its spatial heterogeneity. From 2000 to 2022, 9.4% of the regions experienced significant changes in both NDVI and its spatial heterogeneity. Among them, areas with improving and regrowing conditions, slight, medium, and severe degradation, and desertification, accounted for 62.1%, 11.0%, 16.2%, 8.4%, 2.0%, and 0.3%, respectively. Meadow areas demonstrated a higher proportion of improved conditions and medium degradation, whereas steppe regions had a greater percentage of regrowth and slight degradation. Climatic factors, particularly temperature, have been identified as the primary drivers of vegetation variability within the study area, with a more pronounced impact in meadow regions. Although the importance of human factors in vegetation variation is relatively low, Graz emerged as the predominant driver of NDVI change in severely degraded steppe and meadow regions, with an importance value exceeding 0.50. Notably, the influence of this factor escalated as degradation progressed from slight to severe.

Overall, our results indicate that incorporating the spatial heterogeneity of NDVI into the assessment of vegetation changes can facilitate the detection of early degradation signals along with vegetation greening and can also detect more accurate changes in sparse vegetation areas. Concurrently, this study is capable of highlighting the differences in dominant factors across various subtle types of vegetation change. In the future, a combination of field observations and high-resolution remote sensing data may be needed to deeply explore the causes of these changes in order to better serve ecosystem stewardship.

## Figures and Tables

**Figure 1 plants-13-02814-f001:**
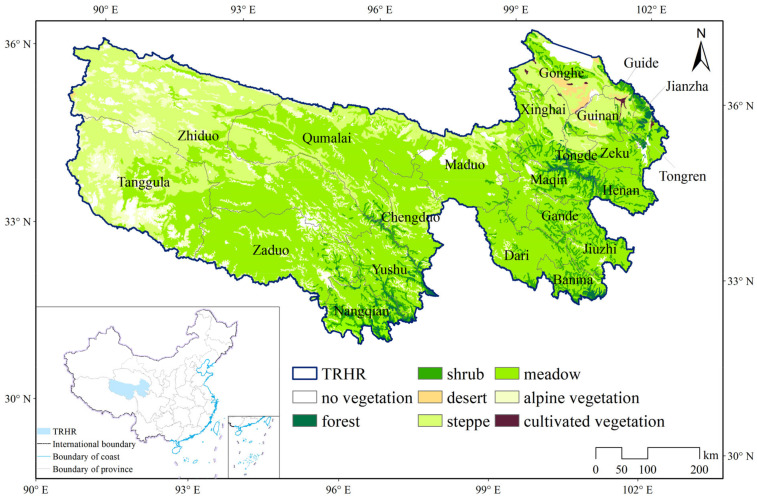
Study area and vegetation types.

**Figure 2 plants-13-02814-f002:**
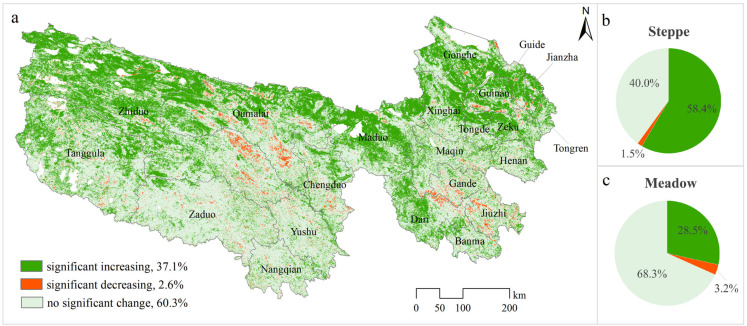
The linear trend in NDVI from 2000 to 2022 in TRHR. ((**a**) Spatial distribution of significant increase, decrease, and no significant change; (**b**,**c**) the respective proportions within steppe and meadow regions).

**Figure 3 plants-13-02814-f003:**
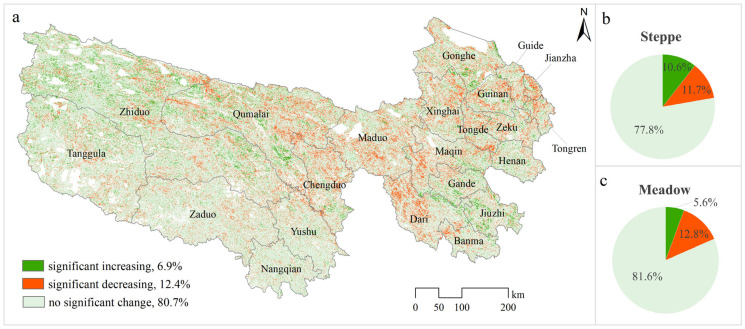
The linear trend in NDVI’s spatial heterogeneity from 2000 to 2022 in TRHR. ((**a**) Spatial distribution of significant increase, decrease, and no significant change; (**b**,**c**) the respective proportions within steppe and meadow regions).

**Figure 4 plants-13-02814-f004:**
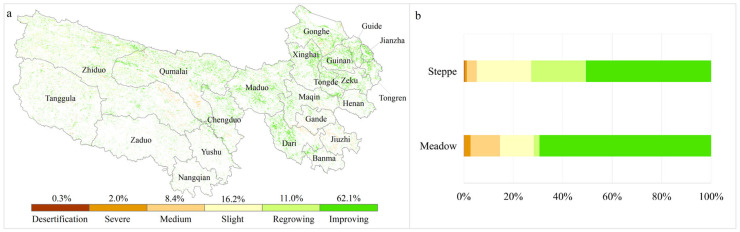
A refined classification of vegetation improvement and degradation from 2000 to 2022 in TRHR. ((**a**) Spatial distribution of six vegetation change categories; (**b**) the respective proportions within steppe and meadow regions).

**Figure 5 plants-13-02814-f005:**
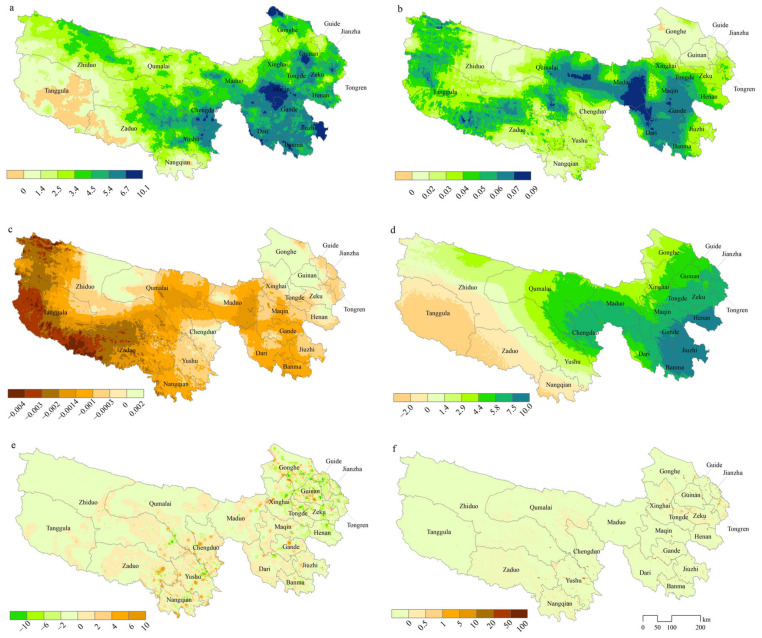
The slope of different driving factors over the past two decades in TRHR (where (**a**–**f**) represent annual precipitation, mean temperature, relative humidity, SPEI–12, grazing intensity, and population density, respectively).

**Figure 6 plants-13-02814-f006:**
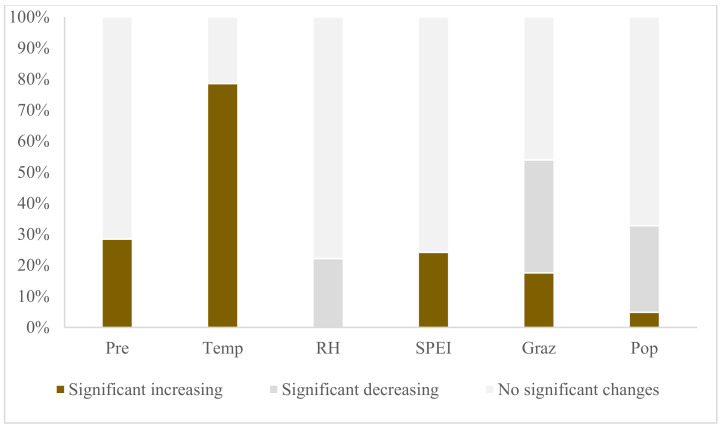
Proportional distribution of variations in different driving factors (%) (Pre, Temp, RH, SPEI, Graz, Pop represent annual precipitation, average temperature, relative humidity, Standardized Precipitation Evapotranspiration Index-12, grazing intensity, and population density, respectively).

**Figure 7 plants-13-02814-f007:**
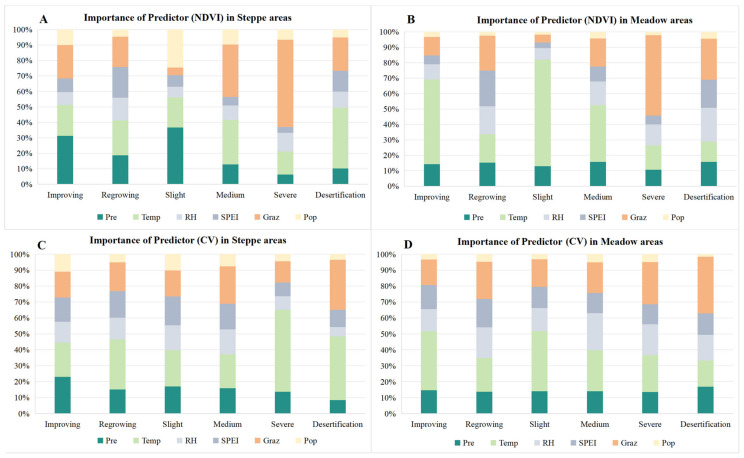
The importance of influencing factors on NDVI and its spatial heterogeneity within different types of significant change areas. ((**A**,**B**) the importance of factors affecting NDVI in steppe and meadow areas, respectively; (**C**,**D**) the importance of factors affecting spatial heterogeneity in these areas).

**Table 1 plants-13-02814-t001:** Classification standards for improvement and degradation.

Types of Ecosystem Change	Required Conditions
Desertification	NDVI < 0.2 and NDVI slope < 0 and CV slope < 0
Severe degradation	NDVI > 0.2 and NDVI slope < 0 and CV slope < 0
Medium degradation	NDVI slope < 0 and CV slope > 0
Slight degradation	NDVI > 0.2 and NDVI slope > 0 and CV slope > 0
Regrowing conditions	NDVI < 0.2 and NDVI slope > 0 and CV slope > 0
Improving conditions	NDVI slope > 0 and CV slope < 0

**Table 2 plants-13-02814-t002:** The proportion of significant, positive, or negative correlation coefficients of different factors with NDVI and its spatial heterogeneity (CV for short in the table).

Name of the Correlation Coefficient	Proportion of Significantly Correlated Regions (%)	Proportion of Positively or Negatively Correlated Areas in All Significantly Correlated Regions (%)
NDVI-Pre	40.1	98.9(+)
NDVI-Temp	27.1	94.6(+)
NDVI-RH	8.2	66.0(−)
NDVI-SPEI	36.0	99.2(+)
NDVI-Graz	24.6	71.5(−)
NDVI-Pop	16.6	84.3(−)
CV-Pre	18.4	82.0(−)
CV-Temp	17.5	61.7(−)
CV-RH	6.6	52.7(−)
CV-SPEI	15.2	87.8(−)
CV-Graz	22.5	50.4(+)
CV-Pop	12.2	66.0(+)

## Data Availability

Normalized Difference Vegetation Index (NDVI), meteorological data, and actual livestock carrying capacity are sourced from National Tibetan Plateau Data Center (https://data.tpdc.ac.cn/zh-hans/data, (accessed on 3 August 2024)). SPEI-12 data are accessible from Zenodo (https://doi.org/10.5281/zenodo.8312201, (accessed on 3 August 2024)). Vegetation types data are from Big Earth Data Center (CAS) (https://data.casearth.cn, (accessed on 2 August 2024)). Land use data are from Resource and Environmental Science Data Platform (http://www.resdc.cn, (accessed on 2 August 2024)). Road network data are sourced from OpenStreetMap (https://www.openstreetmap.org/, (accessed on 20 May 2024)). River network data are obtained from National Cryosphere Desert Data Center (www.ncdc.ac.cn, (accessed on 2 August 2024)). Population density data are from the LandScan Global Population Dataset (Oak Ridge National Laboratory, https://landscan.ornl.gov/, (accessed on 3 August 2024)).

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
