# Peer review of "An Assessment of Vegetation Changes in the Three-River Headwaters Region, China: Integrating NDVI and Its Spatial Heterogeneity"

_plants, 2024, doi:10.3390/plants13192814_

Round 1

Reviewer 1 Report

Comments and Suggestions for Authors

In this study, entitled “Assessment of Vegetation changes in the Three-River Headwaters Region, China: Integrating NDVI and its Spatial Heterogeneity”, the authors assess long-term vegetation dynamics, its spatial heterogeneity, and the potential underlying causes based on a 22-year time series of satellite imagery and NDVI data. Overall, the study is interesting and has some ecological relevance. However, I also have several concerns.

One of my main concerns is how spatial heterogeneity is introduced in the study. The authors claim that changes in vegetation patch-size distribution can be used as an early warning signal of degradation in arid ecosystems. While this is accurate, in this study the authors actually measure how NDVI changes at a minimum scale of 250 m within a moving window of 3x3 pixels. My understanding is that at these working resolutions, changes in NDVI mainly reflect the spatial structure of the amount of vegetation/bare soil cover, rather than patch heterogeneity. Moreover, one of the three objectives of the study is to analyze the main drivers of vegetation dynamics and spatial heterogeneity. However, nothing is mentioned in the introduction regarding this objective.

I appreciate how changes in vegetation dynamics and spatial heterogeneity are combined to inform different types of ecosystems change (Table 1). However, it is imperative to provide more information on the ecological significance of each category so that the reader can understand, for example, why an increase in NDVI can be interpreted as slight degradation. Please, clarify. Also, I find it surprising that there are no regions where the vegetation has not changed. How are pixels classified when NDVI shows a significant temporal trend but no significant changes in CV (and vice versa)? Only those pixels with significant changes in both indicators should be considered. Finally, the results might be more meaningful if reported based on grassland types instead of NDVI ranges. Note that the NDVI distribution in the study area largely corresponds to different grassland communities.

Similarly, the results could be improved with respect to the factors underlying changes in NDVI. For instance, it is unclear to me whether the influence of these factors is evaluated over the trends or for a specific year. Please, clarify. In addition, the direction of the effects is missing. Are they positive or negative effects? In this sense, regarding the predictors of NDVI (Fig. 12), temperature and precipitation are good predictors for both, those pixels that show improvement and those that show degradation. But the effect should be the opposite, shouldn’t it? Further, I think that the importance of each factor could be mapped over the study area, or alternatively, the driving factors could be analyzed based on grassland types, if the authors choose to follow my suggestions.

Additional Comments:

Abstract

L10 Sometimes authors refer to alpine grasslands, sometimes to arid ecosystems. Please standardize.

L21 Add a space before parentheses. Revise throughout the whole manuscript.

Introduction

L36-38 Further elaborate on how increased spatial heterogeneity is derived from a decline in vegetation cover.

L51 Abbreviation is not previously defined.

Methods

L85 The study area seems huge. Would it not be better showing the spatial distribution of characteristics of the study area, rather than providing simple averages? By the way, mean elevation is missing. Also, add a scale bar in Fig. 1

Fig. 1 What is cultural vegetation?

L132-133 Why did the authors choose linear regression over nonparametric analyses such as Theil-Sens’s slope?

Results

L175 This subsection is not directly related to your objectives. Please, consider removing it and begging the results in 3.2

Fig. 4 This figure is quite redundant with Figs. 2 and 3. Please, consider moving it to supplementary material.

Fig. 5 What is the exact meaning of “slight” here?

Fig. 11 Would the authors want to keep the color scheme in this figure?

Discussion

Overall, I find that the discussion focuses too much on repeating the findings of the study, while not emphasizing enough the improvements over previous research and the relevance for the management.

L356 Dynamics

L357-358 This statement is controversial because all arid ecosystems are characterized by patches of vegetation in a matrix of bare soil.

L357-403 It is not clear what the benefits of integrating the spatial heterogeneity are beyond the different estimates. Likely, this could only be achieved with field data.

L459-460 According to Table 1, an increase in NDVI and a decrease in CV is an improvement. Please, clarify.

Comments on the Quality of English Language

Although I am not a native English speaker, I find that the manuscript could greatly benefit from professional English editing and proofreading. There are several places in the text where the wording is ambiguous or sentences could be simplified for greater clarity.

Author Response

Dear reviewer,

Thank you for the constructive and thoughtful comments. The revisions what we made have been shown with revision mode in the manuscript. The responses to the queries are as follows.

  1. Comment: [One of my main concerns is how spatial heterogeneity is introduced in the study. The authors claimthat changes in vegetation patch-size distribution can be used as an early warning signal of degradation in arid ecosystems. While this is accurate, in this study the authors actually measure how NDVI changes at a minimum scale of 250 m within a moving window of 3x3 pixels. My understanding is that at these working resolutions, changes in NDVI mainly reflect the spatial structure of the amount of vegetation/bare soil cover, rather than patch heterogeneity. Moreover, one of the three objectives of the study is to analyze the main drivers of vegetation dynamics and spatial heterogeneity. However, nothing is mentioned in the introduction regarding this objective.]

Our response: Thank you for your valuable comment, we totally agree with your suggestion. Therefore, we have rewritten and introduced the content of patch heterogeneity in the introduction section, please see the detail in lines 39-51. Additionally, the literature on dynamic drivers of vegetation is supplemented in the introduction. The updated section can be found in lines 71-72.

  1. Comment: [I appreciate how changes in vegetation dynamics and spatial heterogeneity are combined to inform different types of ecosystems change (Table 1). However, it is imperative to provide more information on the ecological significance of each category so that the reader can understand, for example, why an increase in NDVI can be interpreted as slight degradation. Please, clarify. Also, I find it surprising that there are no regions where the vegetation has not changed. How are pixels classified when NDVI shows a significant temporal trend but no significant changes in CV (and vice versa)? Only those pixels with significant changes in both indicators should be considered. Finally, the results might be more meaningful if reported based on grassland types instead of NDVI ranges. Note that the NDVI distribution in the study area largely corresponds to different grassland communities.]

Our response: These are excellent suggestions, which we had previously overlooked.

 (1) We have further elaborated on the detailed meanings and explanations of different vegetation change categories in section 2.3.2. Please find the revised content in lines 146-198.

(2) We have revised Fig. 4 and Fig. 5 of the original manuscript (Fig. 2 and Fig. 3 in the revised manuscript), reclassifying the original categories of "slight increase" and "slight decrease" as "no significant change", please see the revisions made at lines 236-237 and lines 250-251. 

(3) Indeed, it makes more sense to report the results based on grassland types, so we have re-conducted the statistics according to the types of steppe and meadow, please see the details in lines 233-235, lines 246-249, and lines 275-281.

  1. Comment: [Similarly, the results could be improved with respect to the factors underlying changes in NDVI. For instance, it is unclear to me whether the influence of these factors is evaluated over the trends or for a specific year. Please, clarify. In addition, the direction of the effects is missing. Are they positive or negative effects? In this sense, regarding the predictors of NDVI (Fig. 12), temperature and precipitation are good predictors for both, those pixels that show improvement and those that show degradation. But the effect should be the opposite, shouldn’t it? Further, I think that the importance of each factor could be mapped over the study area, or alternatively, the driving factors could be analyzed based on grassland types, if the authors choose to follow my suggestions].

Our response(1) We didn't make that clear in our previous manuscript. The influence of these factors is evaluated based on data over the past two decades. We have extracted the multi-year values of NDVI, CV and their influence factors based on the 12 types of pixel points with significant changes and then ran the Random Forest model respectively. A supplementary explanation has been added in section 2.3.3. The revised content can be found in lines 208-210.

(2) To reveal the direction of different influencing factors, we have calculated the Pearson correlation coefficients between NDVI, CV, and each influencing factor for each pixel, and statistically analyzed the proportion of areas that passed the significance test (P<0.05), as well as the proportion of significant positive and negative correlations. For the specific content, please refer to section 3.2.2, lines 318-360. At the same time, the spatial distribution characteristics of NDVI and CV that are significantly correlated with each influencing factor are shown in Supplementary Material's Fig.S2.

(3) It is very important to map the importance of each factor to the study area. However, considering the large scope of the study area, and vegetation dynamics of a single pixel may be affected by a variety of factors within a certain area around it, not just the factors within the pixel itself. Therefore, combining the grassland types and six types of vegetation dynamics divided in 2.3.2, we categorized the significant change areas into 12 distinct types for the driving factors analysis, aiming to discern the varying influences of contributing factors across these regions. Please see the details in section 3.2.3, lines 361-409.

4.Comment: [L10 Sometimes authors refer to alpine grasslands, sometimes to arid ecosystems. Please standardize.]

Our response: We have standardized the terminology, adopting the phrase "arid ecosystem" for consistency. Please see the revision in lines 12 and 32.

5.Comment: [L21 Add a space before parentheses. Revise throughout the whole manuscript.

Introduction]

Our response: We have added a space before parentheses and carefully revised the entire manuscript.

6.Comment: [L36-38 Further elaborate on how increased spatial heterogeneity is derived from a decline in vegetation cover].

Our response: We have provided further explanations in the Introduction, including details of changes in vegetation, bare soil, and landscape during the degradation and improvement of arid ecosystems. Please see the revision in lines 39-51.

7.Comment: [L51 Abbreviation is not previously defined.]

Our response: We have added a definition for the abbreviation TRHR in the abstract. The changes can be found in line 17. 

8.Comment: L85 The study area seems huge. Would it not be better showing the spatial distribution of characteristics of the study area, rather than providing simple averages? By the way, mean elevation is missing. Also, add a scale bar in Fig. 1. Fig. 1 What is cultural vegetation?

Our response: We have supplemented the range of multi-year average temperature, precipitation, and elevation of our study area, the updated content can be found in lines 101-106. Besides, the spatial distribution characteristics of these three factors have been illustrated in Appendix Figure 1. Also we have added a scale bar in Fig.1 and modified “cultural vegetation” to “cultivated plants”, please see the detail in lines 107-108.

9.Comment: L132-133 Why did the authors choose linear regression over nonparametric analyses such as Theil-Sens’s slope?

Our response: Thank you for your valuable suggestion.Considering that linear regression is a commonly used method for analyzing vegetation dynamics, we have employed this approach to analyze the vegetation dynamics of the study area as well as trends of different influencing factors to maintain consistency.

10.Comment: L175 This subsection is not directly related to your objectives. Please, consider removing it and begging the results in 3.2. Fig. 4 This figure is quite redundant with Figs. 2 and 3. Please, consider moving it to supplementary material.

Our response: We have removed the content of Section 3.1 from the original manuscript to streamline the presentation and focus on the core aspects of our study.

11.Comment: Fig. 5 What is the exact meaning of “slight” here?

Our response: The term "slight increase or slight decrease" actually refers to areas that fail the significance test in the original manuscript. To clarify, we have categorized these as "no significant change" in Fig. 2, Fig.3 in the revised manuscript. The revision can be found in lines 236-237 and lines 250-251.

12.Comment: Fig. 11 Would the authors want to keep the color scheme in this figure?

Our response: Following your suggestion, we will retain this color scheme in the figure. Furthermore, we have classified both the areas with slight changes, which did not pass the significance test, and areas with no changes, as “no significant change”. Please see the detail in lines 313-315.

13.Comment: Discussion.Overall, I find that the discussion focuses too much on repeating the findings of the study, while not emphasizing enough the improvements over previous research and the relevance for the management.

Our response: Incorporating the newly calculated correlation coefficients and the updated analysis of driving factors, we have meticulously revised the discussion section. This revision not only enhances and expands upon the improvements over previous research but also underscores the significance of these findings for informed ecosystem management and decision-making. Please refer to lines 411-477 for the revised content.

14.Comment: L356 Dynamics

Our response: Thank you for pointing out the error in the use of singular and plural forms.We have corrected the mistake to ensure the correct use of singular and plural nouns throughout the manuscript. Please see the revision in line 411.

15.Comment: L357-358 This statement is controversial because all arid ecosystems are characterized by patches of vegetation in a matrix of bare soil.

Our response: We apologize for the lack of clarity in our expression, which has led to controversy. We have corrected it and deleted “degraded”. Additionally, following the suggestion of the second reviewer, this section of the content has been moved to the Introduction. Meanwhile, we have supplemented the characteristics of different stages in the degradation and restoration processes of arid ecosystems. Please refer to lines 39-51 for the revised content.

16.Comment: L357-403 It is not clear what the benefits of integrating the spatial heterogeneity are beyond the different estimates. Likely, this could only be achieved with field data.

Our response: Thank you for your comments. In the discussion, we further supplemented the benefits of integrating spatial heterogeneity into the analysis, please see the detail in lines 411-442. We also agree that the benefits of integrating spatial heterogeneity could indeed require further validation through additional field observations. According to the study by Li et al.(2020), the mosaic and distribution of grassland vegetation and bare soil at different stages of degradation have been preliminarily observed during field surveys. However, the spatial heterogeneity estimated by remote sensing data is indeed different from that observed in the field. The lack of field observation data for validation may be one of the limitations in this paper, which we have also emphasized in the discussion, please see the revision in lines 479-483. This still needs to be further deepened in subsequent research.

17.Comment: L459-460 According to Table 1, an increase in NDVI and a decrease in CV is an improvement. Please, clarify.

Our response: Our apologies for the misstatement; we have corrected it to “where NDVI and spatial heterogeneity both increased”. The updated content can be found in line 424. 

  1. Comments on the Quality of English Language:[Although I am not a native English speaker, I find that the manuscript could greatly benefit from professional English editing and proofreading. There are several places in the text where the wording is ambiguous or sentences could be simplified for greater clarity.]

Our response: Thank you for your constructive suggestion. We have carefully revised the English throughout the entire manuscript to make the expression clearer and more fluent.

We believe that these revisions have significantly improved the quality and clarity of our manuscript. We appreciate the reviewers' insights and are grateful for the opportunity to enhance our work.

Sincerely,

Xuejie Mou

Institute of Ecological Protection and Restoration Planning, Yellow River Ecological Civilization Research Center

Chinese Academy of Environmental Planning

Reviewer 2 Report

Comments and Suggestions for Authors

The manuscript "Assessing Vegetation Change in the Three-River Headwaters Region, China: Integrating NDVI and its Spatial Heterogeneity" by Mou et al. addresses a topical issue and is well written. However, my comments in detail are below: 

The introduction should better bring the reader into the topic. At the same time, previous studies are not well delimited.

Materials and Methods: lines 92-94: references are missing.

101, 110: MODIS, SPEI-explain the abbreviations.

120-122: You should not refer to previous studies in this section.

158-160: delete this phrase. It would be best if you did not refer to previous studies in this section your study, only.

357-362: delete this paragraph or move it to the introduction. It does not belong here.

Author Response

Dear reviewer,

Thank you for the constructive comments. The revisions what we made have been shown with revision mode in the manuscript. The responses to the queries are as follows.

1.Comment: [The introduction should better bring the reader into the topic. At the same time, previous studies are not well delimited.]

Our response:We greatly appreciate your suggestion. We have reorganized the logic of the Introduction, added the latest research progress related to our study area, and pointed out the gaps in current research. Then we have further clarified the efforts made by this study to fill these gaps. Please see the revised content in lines 39-91.  

2.Comment: [Materials and Methods: lines 92-94: references are missing.]

Our response:We have supplemented the relevant references about the climate characteristics of the study area. The revision can be found in line 101.

3.Comment: [101, 110: MODIS, SPEI-explain the abbreviations.]

Our response:We have added the full names of MODIS and SPEI in the revised manuscript. Please refer to lines 113-114 and 123 for details.

4.Comment: [120-122: You should not refer to previous studies in this section.]

Our response:We have deleted these content. Please see the revision in line 133.

5.Comment: [158-160: delete this phrase. It would be best if you did not refer to previous studies in this section your study, only.]

Our response:We have removed this phrase. The modification can be found in line 201.

6.Comment: [357-362: delete this paragraph or move it to the introduction. It does not belong here.]

Our response:We have removed this paragraph. Please see line 412 for the revision.

We believe that these revisions have significantly improved the quality and clarity of our manuscript. We appreciate the reviewers' insights and are grateful for the opportunity to enhance our work.

Sincerely,

Xuejie Mou

Institute of Ecological Protection and Restoration Planning, Yellow River Ecological Civilization Research Center

Chinese Academy of Environmental Planning

Round 2

Reviewer 1 Report

Comments and Suggestions for Authors

In this second version, the authors have done an excellent job and have clearly improved the manuscript by addressing most of the suggestions and conducting a critical analysis or their own work. Furthermore, the manuscript has gained in clarity and readability after English language revisions. However, there are still some aspects that need further attention and clarification.

I agree with the authors’ decision to focus the study on comparing the two types of grasslands in the region: meadows vs. steppe. However, these types of grasslands should be introduced to the reader, which is not currently done in the manuscript. For instance, what are the characteristics of these ecosystems? What is their ecological importance? Moreover, Fig. 1 does not depict these grasslands types; instead, it shows a broader variety of grassland types. By including this information, perhaps in the “study area” section of the methods, would greatly enhance the understanding of the ecological meaning of the six vegetation change typologies.

Additionally, I still don’t fully understand how the random forest model is used. Essential information is still missing. For instance, it is unclear to me which is the response variable used. Are the NDVI and CV time series per pixel used, or are the final trend values used? Please, clarify. Also, I am not totally convinced of the convenience of the correlation analysis. Can’t the random forest analysis itself provide coefficients and direction of change? This is a crucial part of the study and must be explained with utmost clarity. Along these lines, I find that the discussion is still too focused on repeating the results. The author could, for instance, explore specific management measures considering the most important factors driving vegetation changes. For example, how can degradation be mitigated in the affected areas? When the livestock is involved, one might suggest changes in grazing management practices, such as reducing livestock stocking rates. But what can be done when temperature and precipitation are the key variables?

Minor comments:

L12 “alpine arid and fragile ecosystems”

L165 “gr” - check for a potential typo

L251 What does this 9.4% represents in term of land surface (in ha)? It seems a little fraction of the study area, but I would say that it is an extensive territory to effectively be managed

L262 “es16.2%” -check for potential typo. Please, thoroughly revise the whole manuscript for similar typos

Fig. 6 The background of the figure should perhaps be white

Fig. 7 Rename panels from a1-a1 to A-D

Author Response

Dear Reviewer,

Thank you for the constructive and thoughtful comments. The revisions what we made have been shown with revision mode in the manuscript. The responses to the queries are as follows.

  1. Comment:[In this second version, the authors have done an excellent job and have clearly improved the manuscript by addressing most of the suggestions and conducting a critical analysis or their own work. Furthermore, the manuscript has gained in clarity and readability after English language revisions. However, there are still some aspects that need further attention and clarification.I agree with the authors’ decision to focus the study on comparing the two types of grasslands in the region: meadows vs. steppe. However, these types of grasslands should be introduced to the reader, which is not currently done in the manuscript. For instance, what are the characteristics of these ecosystems? What is their ecological importance? Moreover, Fig. 1 does not depict these grasslands types; instead, it shows a broader variety of grassland types. By including this information, perhaps in the “study area” section of the methods, would greatly enhance the understanding of the ecological meaning of the six vegetation change typologies.]

Our response: Thank you very much for the valuable suggestion. We neglected this in the last round of revisions. We have added a description of the vegetation types and their proportions in the study area as shown in Figure 1. Additionally, we have also supplemented the main characteristics and significance of meadows and steppes as the dominant ecosystem types. Please refer to lines 97-107. The legend "shrub" in Figure 1 has been corrected, please see the revision in line 114.

  1. Comment:[Additionally, I still don’t fully understand how the random forest model is used. Essential information is still missing. For instance, it is unclear to me which is the response variable used. Are the NDVI and CV time series per pixel used, or are the final trend values used? Please, clarify. Also, I am not totally convinced of the convenience of the correlation analysis. Can’t the random forest analysis itself provide coefficients and direction of change? This is a crucial part of the study and must be explained with utmost clarity. Along these lines, I find that the discussion is still too focused on repeating the results. The author could, for instance, explore specific management measures considering the most important factors driving vegetation changes. For example, how can degradation be mitigated in the affected areas? When the livestock is involved, one might suggest changes in grazing management practices, such as reducing livestock stocking rates. But what can be done when temperature and precipitation are the key variables?]

Our response: Thank you for your advice. (1) In this study, NDVI and its spatial heterogeneity (CV) were taken as dependent variables respectively, which has been mentioned in section 2.3.3, Please refer to lines 213-214. We actually used the time series of NDVI and CV for each pixel to perform the Random Forest model calculations. For details of the specific calculation process, please refer to lines 217-222.

(2) Random Forest (RF) is an ensemble learning method that operates by constructing a multitude of decision trees during training time and outputting the mode of the classes (classification) or averaging the predictions (regression) of the individual trees to form a prediction. It is particularly useful in driver factor analysis because it can handle a large number of input variables and provide measures of variable importance, which can help identify the key factors driving changes in a system. However, it does not output correlation coefficients among the variables. This is because Random Forest is an ensemble method that averages the predictions of multiple decision trees, and its primary focus is on prediction accuracy rather than measuring linear relationships between variables. Therefore, we primarily calculated the correlation coefficients to explore the direction of the factors' impact on NDVI and CV.

(3) To emphasize the significance of our research findings for guiding ecosystem management, we have added relevant content to section 4.2, retitling it 'Influencing factors of vegetation changes and suggestions for ecosystem management'. For specifics, please refer to lines 453, 480-485, and 497-507. Additionally, we have updated the analysis of climate and anthropogenic trends, as well as the extent of their impacts, in Section 4.2, Paragraph 1, please see the detail in lines 457-464.

  1. Comment: [L12 “alpine arid and fragile ecosystems”]

Our response: Thank you for your suggestion.We have revised it in line 12.

  1. Comment: [L165 “gr” - check for a potential typo]

Our response: Please accept our apologies for the oversight. It appears that some content was inadvertently removed during the manuscript revision process, and we have since restored it. For details, please see lines 173-177. We have also checked the entire manuscript to avoid similar situations.

  1. Comment:[L251 What does this 9.4% represents in term of land surface (in ha)? It seems a little fraction of the study area, but I would say that it is an extensive territory to effectively be managed]

Our response: We acknowledge that while 9.4% may not seem like a significant proportion, it actually corresponds to an area of approximately 36,096 square kilometers, which is indeed quite substantial. To provide a more precise understanding, we have included the actual area size of 9.4% in lines 264-265 of our manuscript.

  1. Comment:[L262 “es16.2%” -check for potential typo. Please, thoroughly revise the whole manuscript for similar typos]

Our response: Apologies for the typo, we have made the correction, changing it to "approximately", please see the revision in line 277. We have also checked the entire manuscript to avoid similar situations.

  1. Comment:[Fig. 6 The background of the figure should perhaps be white]

Our response: This could be a change that occurred after converting to PDF; we have set the background of Figure 6 to white. Please see the revision in line 323.

  1. Comment:[Fig. 7 Rename panels from a1-a4to A-D]

Our response: Thank you for your suggestion. We have revised Figure 7 according to your feedback. Please refer to line 417.

Best regards,

Xuejie Mou

Institute of Ecological Protection and Restoration Planning, Yellow River Ecological Civilization Research Center

Chinese Academy of Environmental Planning

Reviewer 2 Report

Comments and Suggestions for Authors

The authors have made efforts to respond to the reviewer's comments. I consider that the manuscript complies with the standards for publication.

Author Response

Dear Reviewer,

Thank you very much for your positive evaluation and for the time and effort you have dedicated to reviewing our manuscript. We are grateful for your valuable feedback and are pleased that you find our work meets the standards for publication. We are looking forward to the next steps in the publication process.

Once again, thank you for your support and for your contribution to the quality of our research.

Best regards,

Xuejie Mou

Institute of Ecological Protection and Restoration Planning, Yellow River Ecological Civilization Research Center

Chinese Academy of Environmental Planning
